# A Metalearned Neural Circuit for Nonparametric Bayesian Inference

**Jake C. Snell**
Department of Computer Science
Princeton University
jsnell@princeton.edu

**Gianluca M. Bencomo**
Department of Computer Science
Princeton University
gb5435@princeton.edu

**Thomas L. Griffiths**
Department of Computer Science
Department of Psychology
Princeton University
tomg@princeton.edu

## Abstract

Most applications of machine learning to classification assume a closed set of balanced classes. This is at odds with the real world, where class occurrence statistics often follow a long-tailed power-law distribution, rarely revealing the entire problem domain in a single sample. Nonparametric Bayesian models naturally capture this phenomenon, but have significant practical barriers to widespread adoption, namely implementation complexity and computational inefficiency. To address this, we present a method for extracting the inductive bias from a nonparametric Bayesian model and transferring it to an artificial neural network. By simulating data with a nonparametric Bayesian prior, we can metalearn a sequence model that performs inference over an unlimited set of classes. After training, this "neural circuit" has distilled the corresponding inductive bias and can successfully perform sequential inference over an open set of classes. Our experimental results show that the metalearned neural circuit achieves comparable or better performance than particle filter-based methods that explicitly perform Bayesian nonparametric inference while being faster and simpler to use.

## 1 Introduction

Standard machine learning approaches to classification assume that the set of possible classes is known *a priori*. Classification in this setting thus involves selecting the most appropriate class label from a closed set. However, this is not the case for human learners. Imagine European explorers in Australia seeing a kangaroo for the first time. Rather than trying to classify this observation into an existing class – is it a deer or a rabbit? – they recognized that a new class needs to be created. Although this is easy for humans, current machine learning systems struggle to identify novel classes and use them in predictions [46].

Bayesian statistics offers an elegant solution to the problem of novel classes: define a model in a way that does not make a commitment to an upper bound on the number of classes. This idea is expressed in nonparametric Bayesian models, namely the Dirichlet process mixture model (DPMM) [3]. In this model, a new data point is assumed to be generated from an existing class with probability proportional to the number of previous observations from that class and from a new class with a probability proportional to $\alpha$, a hyperparameter of the model. This makes it possible to both postulate

38th Conference on Neural Information Processing Systems (NeurIPS 2024).

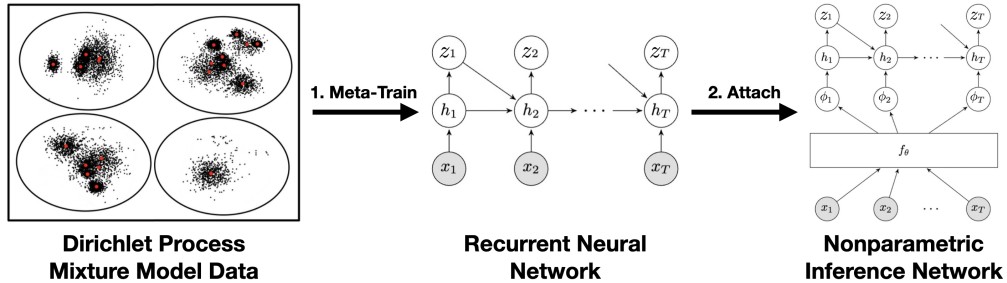

**Dirichlet Process Mixture Model Data**     **Recurrent Neural Network**     **Nonparametric Inference Network**

Figure 1: Our proposed nonparametric inference network first internalizes the desired nonparametric Bayesian prior via metalearning a recurrent neural network (RNN) to model its posterior distribution over class assignments. Afterwards, the metalearned RNN, or *neural circuit*, has captured the corresponding inductive bias and can be used to perform sequential inference over a potentially unbounded number of classes.

that a new datapoint might come from a new class and capture long-tailed distributions of class frequency commonly found in real-world classification problems.

Despite their elegance, nonparametric Bayesian models have fallen out of favor in machine learning, as they are difficult to reconcile with the current focus on large-scale models defined over complex objects such as images. The scalability of non-parametric Bayesian models is limited due to the high computational cost of Bayesian inference, typically requiring the use of sampling algorithms such as Markov chain Monte Carlo [44] or particle filters [23]. Applying these models to complex objects requires creativity in defining generative models that are sufficiently expressive without making inference intractable. Furthermore, most standard Bayesian inference algorithms are not designed to perform sequential inference, instead assuming that all data are presented in a single batch.

In this paper we pursue a different approach to inference in nonparametric Bayesian models: training a recurrent neural network (RNN) to approximate the posterior distribution over classes from a DPMM. We formulate this problem as one of *metalearning*, repeatedly sampling a sequence of class memberships and observations from the DPMM and training the model to predict the class of each observation conditioned on the class labels of those preceding it. The resulting *neural circuit* (Figure 1) internalizes the inductive bias of the nonparametric Bayesian model used to generate the training tasks. It can then be used as a component in deep neural networks, extending nonparametric Bayesian methods to complex objects such as images.

Our approach combines the elegance of Bayesian nonparametric models with the predictive power of deep learning in a principled and practical manner. The discriminative nature of the RNN allows it to classify complex inputs without making the restrictive distributional assumptions required by standard Bayesian inference algorithms. Since RNNs are sequence models, the neural circuit can efficiently make predictions for a sequence of observations as each successive predictive distribution is computed in constant time. We apply our neural circuit to a challenging open-set image classification tasks from the ImageNet [51] and iNaturalist [61] datasets and show it achieves excellent predictive performance at a fraction of the computational cost of standard DPMM inference algorithms.

## 2 Background

Consider the task of a learner classifying objects in a new environment. The learner encounters a sequence of items $\mathbf{x}_1, \mathbf{x}_2, \ldots, \mathbf{x}_T$. After an item $\mathbf{x}_t$ is observed, the learner attempts to predict its class label. The learner is then presented with the true class label $z_t$, updates internal representations as necessary, and repeats the procedure for timestep $t + 1$. The predictive distribution over the entire sequence of labels $z_{1:T} = z_1, \ldots, z_T$ can be written as

$$p(z_{1:T}|\mathbf{x}_{1:T}) = p(z_1|\mathbf{x}_1) \prod_{t=2}^{T} p(z_t|\mathbf{x}_{1:t}, z_{1:t-1}). \tag{1}$$

We first review Dirichlet process mixture models, which provide a framework for expressing (1) as posterior inference over $z_{1:T}$ without restricting the labels to belong to a closed set. We then

review the particle filter of Fearnhead [23], which provides a method for sequentially computing the predictive distribution through the use of weighted particles representing assignments of class labels.

## 2.1 Dirichlet Process Mixture Models

The nonparametric Bayesian solution to the prediction task presented above involves infinite mixture models, the most notable of which is the Dirichlet process mixture model (DPMM) [3, 22, 47]. There are two components to the DPMM: a distribution over class memberships and a class-conditional distribution over observations. In this model, observations are generated from latent classes, where the distribution over classes places no limit on the number of classes. The joint probability follows a Markov process on $z_{1:T}$ and assumes class-conditional independence for $\mathbf{x}_{1:T}$:

$$p(z_{1:T}, \mathbf{x}_{1:T}) = p(z_1)p(\mathbf{x}_1|z_1) \prod_{t=2}^{T} p(z_t|z_{1:t-1})p(\mathbf{x}_t|z_t). \tag{2}$$

The conditional distribution on class memberships $p(z_t|z_{1:t-1})$ is relatively simple:

$$p(z_t = k \mid z_{1:t-1}) \propto \begin{cases} n_k & k \text{ previously observed} \\ \alpha & k \text{ is a new class,} \end{cases} \tag{3}$$

where $n_k$ denotes the number of occurrences of class $k$ in $z_1, \ldots, z_{t-1}$, and by convention the value of $k$ for a new class is taken to be one greater than the number of classes observed so far. This process is known as the Chinese restaurant process (CRP) [1].

The conditional distribution over observations is $p(\mathbf{x}_t \mid z_t = k) = g(\mathbf{x}_t|\boldsymbol{\varphi}_k)$, where $g(\cdot|\boldsymbol{\varphi})$ is some probability function with parameter $\boldsymbol{\varphi}$. Each $\boldsymbol{\varphi}_k$ is in turn distributed according to a shared prior $\pi(\boldsymbol{\varphi}_k)$ for $k = 1, 2, \ldots$. The predictive distribution over class assignments (1) can be derived by a simple application of Bayes' rule to the joint distribution (2).

Direct computation of the posterior in a DPMM is intractable, and therefore several methods have been developed to perform approximate inference using methods such as MCMC [44] and variational inference [9]. However, these methods typically assume that all observations are presented simultaneously and do not attempt to handle the sequential nature inherent to our problem formulation.

## 2.2 Particle Filter for the DPMM

One notable method that *does* aim to perform sequential inference of class labels in the DPMM is the particle filter proposed by Fearnhead [23]. Particle filters maintain a set of weighted particles at each timestep that approximate the posterior distribution over latent variables [14, 19]. At each timestep, a particle filter propagates a set of particles forward in time by first sampling a new set of states from a transition distribution and then assigning weights according to a potential function chosen such that the set of weighted particles approximates the target posterior distribution.

In order to make inference tractable in the DPMM setting, Fearnhead [23] assumes that $g(\mathbf{x}|\boldsymbol{\varphi})$ belongs to the exponential family and $\pi(\boldsymbol{\varphi})$ is the corresponding conjugate prior. In this case, it is possible to marginalize over $\boldsymbol{\varphi}$ when computing the posterior predictive distribution for a class. Suppose that $\mathbf{x}_i \sim g(\mathbf{x}_i|\boldsymbol{\varphi})$ for $i = 1, \ldots, n + 1$. Since the posterior $p(\boldsymbol{\varphi}|x_{1:n})$ has the same form as the conjugate prior $\pi(\boldsymbol{\varphi})$, there exists a closed form expression for the posterior predictive distribution

$$p(\mathbf{x}_{n+1}|\mathbf{x}_{1:n}) = \int p(\mathbf{x}_{n+1}|\boldsymbol{\varphi})p(\boldsymbol{\varphi}|\mathbf{x}_{1:n}) \, d\boldsymbol{\varphi}. \tag{4}$$

Each particle in the method of Fearnhead [23] represents an entire trajectory of class labels up to the current timestep. Suppose that after $t$ timesteps there are $J$ particles $z_{1:t}^{(j)}$ each with weight $w_j \geq 0$ for $j = 1, \ldots, J$ such that $\sum_{j=1}^{J} w_j = 1$. For each particle, the transition distribution is:

$$p(z_{t+1} = k \mid \mathbf{x}_{1:t+1}, z_{1:t}^{(j)}) \propto p(z_{t+1} = k|z_{1:t}^{(j)})p(\mathbf{x}_{t+1}|\mathbf{x}_{1:t}, z_{1:t}^{(j)}, z_{t+1} = k), \tag{5}$$

where the likelihood term $p(\mathbf{x}_{t+1}|\mathbf{x}_{1:t}, z_{1:t}^{(j)}, z_{t+1} = k)$ can be computed in closed form according to (4). If $k$ represents a new class, this will be the prior predictive probability instead.

Two major shortcomings of the particle filter are that it makes restrictive distributional assumptions (the class conditional distribution needs to be exponential family) and requires many particles in order to sufficiently approximate the posterior. In the next section, we present our metalearned neural circuit which is aimed at addressing these issues.

# 3 Metalearning a Neural Circuit

We propose a novel amortized inference approach for class inference in the DPMM, based on metalearning a recurrent neural network (RNN) to predict class memberships. By applying metalearning to tasks defined by sampling data from a DPMM, the RNN can effectively internalize the corresponding inductive bias in a reusable neural network, hence the name *neural circuit*.

Our approach is inspired in part by the observation that the updates of the particle filter in Section 2.2 can be implemented by accumulating sufficient statistics of observations. Recall that an exponential family distribution and its conjugate prior can be expressed in terms of natural parameters $\boldsymbol{\eta}$:

$$p(\mathbf{x} \mid \boldsymbol{\eta}) = h(\mathbf{x}) \exp\left\{ \langle \boldsymbol{\eta}, \mathbf{t}(\mathbf{x}) \rangle - A(\boldsymbol{\eta}) \right\} \tag{6}$$

$$p(\boldsymbol{\eta} \mid \boldsymbol{\tau}, \nu) = \exp\left\{ \langle \boldsymbol{\tau}, \boldsymbol{\eta} \rangle - \nu A(\boldsymbol{\eta}) - B(\boldsymbol{\tau}, \nu) \right\}, \tag{7}$$

where $\mathbf{t}(\mathbf{x})$ are the sufficient statistics and $A(\boldsymbol{\eta})$ is the log normalizer. The posterior after observing $\mathbf{x}_{1:n}$ is of the same form as the prior, namely $p(\boldsymbol{\eta} \mid \boldsymbol{\tau}', \nu')$ where

$$\boldsymbol{\tau}' = \boldsymbol{\tau} + \sum_{i=1}^{n} \mathbf{t}(\mathbf{x}_i) \qquad \nu' = \nu + n. \tag{8}$$

From this perspective, the particle filter can be implemented by first initializing the representation of each class to be $\boldsymbol{\tau}, \nu$. Then the sufficient statistics $\mathbf{t}(\mathbf{x}_t)$ for each observation are extracted and used to update the corresponding class's $\boldsymbol{\tau}$ after the true label is revealed.

An analogous computation is carried out by a recurrent network, which given some input $\mathbf{x}_t$ and previous hidden state $\mathbf{h}_{t-1}$, computes an updated hidden representation $\mathbf{h}_t$ and an output $\mathbf{u}_t$:

$$\mathbf{h}_t, \mathbf{u}_t \leftarrow \mathrm{RNN}_{\boldsymbol{\theta}}(\mathbf{x}_t, \mathbf{h}_{t-1}) \tag{9}$$

The differences with respect to the particle filter are twofold: the representation of each cluster is no longer separate but shared in $\mathbf{h}_t$, and the representations are learnable end-to-end.

We also recognize that the output $\mathbf{u}_t$ can be the basis for predicting the current class label, and $\mathbf{h}_t$ can be made to capture the updated current state of all classes simultaneously. To predict the current class label, the RNN's output $\mathbf{u}_t$ is mapped to a provisional logit $\mathbf{a}_t$ using a learnable weight matrix $\mathbf{W}$ and bias vector $\mathbf{b}$. The logits are then additively masked by $\mathbf{m}_t$, which preserves the logit as long as predicting the corresponding class would be valid (i.e. the class label is at most one greater than any previously seen class label). The input to the RNN at each timestep is the concatenation of the current observation $\mathbf{x}_t$ and a one-hot representation of the previous label $z_{t-1}$ (all zeros when $t = 1$). The predictive distribution is therefore defined to be:

$$p_{\boldsymbol{\gamma}}(z_t | \mathbf{x}_{1:t}, z_{1:t-1}) = \mathrm{SOFTMAX}(\mathbf{a}_t + \mathbf{m}_t) \tag{10}$$

$$\mathbf{a}_t = \mathbf{W}\mathbf{u}_t + \mathbf{b}$$

$$m_{tk} = \begin{cases} 0 & k \leq 1 + \max z_{1:t-1} \\ -\infty & \text{otherwise}, \end{cases}$$

$$\mathbf{u}_t, \mathbf{h}_t \leftarrow \mathrm{RNN}_{\boldsymbol{\theta}}([\mathbf{x}_t, \mathrm{ONE\text{-}HOT}(z_{t-1})], \mathbf{h}_{t-1}),$$

where $\boldsymbol{\gamma} \triangleq \{\boldsymbol{\theta}, \mathbf{W}, \mathbf{b}\}$ are learnable parameters.

In order to learn these weights, we turn to metalearning. In metalearning, a system is presented with a set of tasks sampled from a distribution over tasks. The goal is to leverage the shared structure of these tasks not only to become better at solving each individual task but also to solve future tasks better, effectively "learning to learn" [4, 8, 30, 56]. This is done by estimating a set of hyperparameters shared across tasks. In our case, these are the weights of the neural circuit, which we metalearn by minimizing negative log-likelihood of item-label sequences generated by a DPMM.

Let $z_{1:T}$ be generated according to a CRP (3) and $\mathbf{x}_t \mid z_t = k$ be generated according to some (possibly unknown) distribution $g(\mathbf{x}_t | \boldsymbol{\varphi}_k)$ for $t = 1, \ldots, T$. Let $\mathcal{D}$ denote this joint distribution over $(z_{1:T}, \mathbf{x}_{1:T})$. We define

$$\boldsymbol{\gamma}^* = \arg\min_{\boldsymbol{\gamma}} \mathbb{E}_{(z_{1:T}, \mathbf{x}_{1:T}) \sim \mathcal{D}} \left[ -\frac{1}{T} \sum_{t=1}^{T} \log p_{\boldsymbol{\gamma}}(z_t | \mathbf{x}_{1:t}, z_{1:t-1}) \right], \tag{11}$$

where the minimization is performed via a gradient-based optimization procedure. This metalearning approach can be applied equally well whether the the form of the class-conditional distributions is known or not, since all that is required are samples from the joint distribution. If the class-conditional distribution is unknown, it is taken to be the empirical distribution constructed from a dataset by placing an evenly-weighted point mass on each sample within a class.

**Relationship to Amortized Inference**    Our method can be viewed as a form of amortized inference, wherein a function (e.g. represented by a deep neural network) is learned to directly map from inputs to an approximate posterior distribution over latent variables [35, 49]. Amortized inference is similar to discriminative classification in that it directly maps from inputs to class labels, but it can also be viewed as approximate Bayesian inference within the generative modeling framework. Networks trained to perform amortized inference are sensitive to the choice of prior, since different priors will lead to different posteriors, the KL-divergence to which will be minimized during learning. In a similar fashion, our method is also affected by the choice of DPMM prior as different priors change the distribution over sequence that the neural circuit is trained on.

**Relationship to Metalearning**    A variety of methods have been used to learn shared hyperparameters to solve a set of related tasks problem for deep neural networks [2, 24, 48, 53, 63], but our approach is most similar to those in which a recurrent neural network is trained to perform multiple tasks [20, 65]. This approach is more typically used in reinforcement learning, defining a system that learns a global meta-policy that supports efficient learning on specific tasks. The resulting RNNs have been shown to encode information equivalent to a Bayesian posterior distribution [40], making them a good candidate for metalearning amortized Bayesian inference.

# 4    Experiments

We apply the neural circuit on three data settings: a synthetic dataset where the form of the DPMM is known, sequences of labels generated from a CRP on ImageNet [51], and sequences sampled directly from the long-tailed iNaturalist 2018 species classification dataset [61]. The goal of our experiments is to compare both the predictive performance and computational efficiency of the neural circuit to standard sequential inference techniques for the DPMM.

Our main experimental point of comparison is the particle filter of Fearnhead [23] discussed in Section 2.2. For a non-Bayesian baseline, we compare to a softmax-based classifier that augments the distribution over $K$ known classes at time $t$ with an additional logit representing the possibility of a new class. The logit representing the new class is derived using the maximum entropy principle [33] by introducing the constraint that the marginal probability that $z_t$ is a new class be equal to $\alpha/(t-1+\alpha)$ (please refer to Appendix B for more details). We refer to this baseline as "Softmax + Energy."

We consider two main evaluation scenarios. The first, referred as the *sequential observation setting*, measures the quality of a model's predictions when the true class label is provided after each prediction. We report the average predictive negative log-likelihood (NLL) averaged across timesteps and corresponding perplexity. The second setting quantifies how well the model predicts when there is no feedback about the true class labels (we refer to this as the *fully unobserved setting*). We compare the maximum a posteriori (MAP) prediction of the model against the true sequence of class labels using standard clustering metrics, including the adjusted Rand index (ARI) [31] and adjusted mutual information score (AMI) [62]. In both settings, we compare computational efficiency by evaluating the wall clock compute time per sequence to make predictions. Please refer to Appendix D for additional experimental details.

## 4.1    Modeling Synthetic Data from a DPMM

We first evaluate performance of the neural circuit on a synthetically generated DPMM dataset. Our aim is to determine whether the neural circuit can match performance of the particle filter when the class-conditional distributions belong to the exponential family with a known prior. Specifically, we use a normal-inverse-gamma prior and Gaussian class conditional distributions with unknown mean and variance. For each dimension $d = 1, \ldots, D$, the form of the class-conditional distribution is:

$$\sigma_d^2 \sim \Gamma^{-1}(a, b) \qquad \mu_d \sim \mathcal{N}(m, \sigma_d^2/\lambda) \qquad x_d \sim \mathcal{N}(\mu_d, \sigma_d^2),$$

where $m$, $\lambda$, $a$, and $b$ are known hyperparameters. We set the length of the sequences to $T = 100$ and chose $D = 2$, $m = 0$, $\lambda = 0.01$, and $\alpha = \beta = 2$. Several sequences drawn from this distribution are shown in Figure 2. We directly set the hyperparameters of the particle filter to their true values.

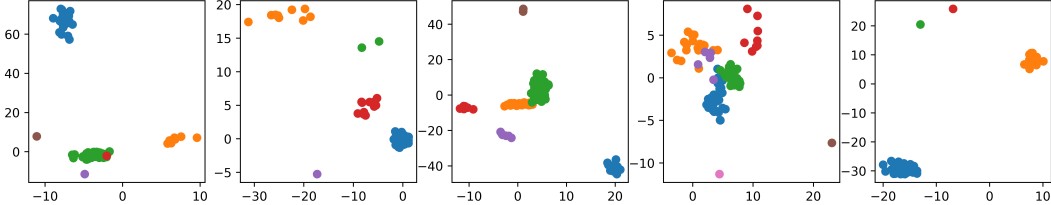

Figure 2: Visualization of sample synthetic sequences generated from the normal-inverse-prior used in Section 4.1. Classes are sampled from a Chinese restaurant process (CRP) with $\alpha = 1.0$ and sequences consist of 100 timesteps. Clusters colored by true class label.

The results of our evaluation can be found in Table 1. The Softmax + Energy baseline is significantly slower than the other methods due to solving for weights after each timestep. Although its NLL is better than the CRP, the average perplexity is worse due to the presence of some sequences where NLL is quite poor. These sequences are accentuated due to exponentiation in the formula for computing perplexity. We find that although the particle filter performs best on negative log-likelihood, the neural circuit provides better clusterings in the fully unobserved setting. We hypothesize this is due to the particle filter's insufficient exploration of the posterior over labelings stemming from its reliance on particles. We also find that neural circuit inference is roughly $5\times$ faster than the particle filter in the sequential observation setting and roughly $10\times$ faster in the fully unobserved setting.

Table 1: Results on two-dimensional data synthesized from a DPMM. Evaluation computed as average over 10,000 held-out sequences of length 100. Negative log-likelihoods are expressed in nats per timestep.

| | Seq. Observation | | Fully Unobserved | | Inference Time (ms/seq.) | |
|---|---|---|---|---|---|---|
| Method | NLL ($\downarrow$) | Perp. ($\downarrow$) | ARI ($\uparrow$) | AMI ($\uparrow$) | Seq. Obs. ($\downarrow$) | Fully Unobs. ($\downarrow$) |
| CRP | 1.006 | 2.978 | 0.010 | 0.010 | **0.019** | **0.179** |
| Softmax + Energy | 0.929 | 24.742 | 0.388 | 0.392 | 1679.716 | 1691.403 |
| Particle Filter | **0.048** | **1.053** | 0.769 | 0.814 | 1.617 | 4.432 |
| Neural Circuit | 0.076 | 1.086 | **0.921** | **0.928** | 0.059 | 0.421 |

## 4.2 Open-set Classification on ImageNet-CRP

Next we consider a challenging open-set image classification task where the input features are activations from a ResNet [29]. Our goal is to determine whether the neural circuit can effectively scale up to a high dimensional space where the form of the class-conditional distribution is unknown.

We downloaded the weights of a pretrained ResNet-18 from TIMM [66] and extracted the 512-dimensional penultimate layer activations from the entire ILSVRC 2012 dataset [51]. We split the 1,000 classes into 500 reserved for training (meta-train classes) and 500 for testing (meta-test classes). We generate sequences by sampling $z_{1:N}$ from a CRP with $\alpha = 1.0$, assigning each distinct value of $z_n$ to a class uniformly at random, and then sampling the corresponding observation $x_n$ uniformly from the images belonging to that class. We call this data-generating procedure ImageNet-CRP.

We metalearn the neural circuit with the same architecture and optimization procedure as in Section 4.1. For the particle filter, additional care needs to be taken to model the sparse nonnegative ResNet features that are produced by a ReLU activation. We modeled this using a hurdle [16] model with log-normal distribution over nonnegative values. This model posits a log-normal distribution over the nonnegative values and a point mass at zero:

$$x_d \sim \tilde{x}_d \cdot y_d \qquad \tilde{x}_d \sim \text{LogNormal}(\mu_d, \sigma_d^2) \qquad y_d \sim \text{Bern}(p_d) \tag{12}$$

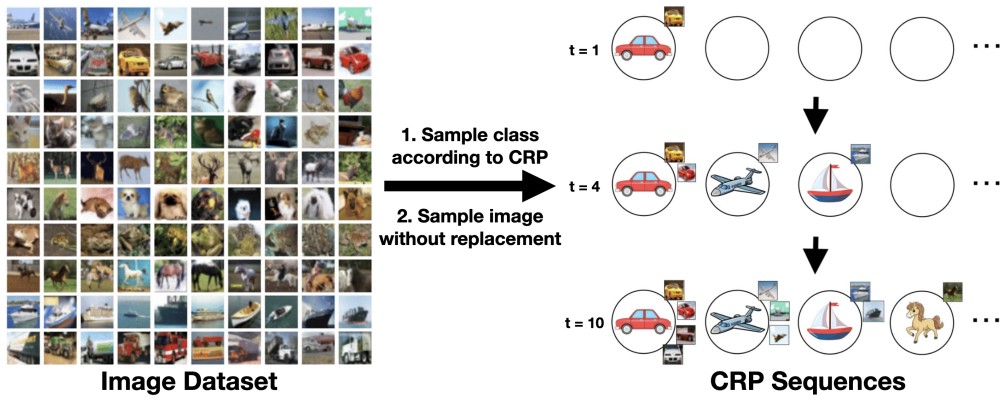

**Image Dataset**          **CRP Sequences**

Figure 3: Diagrammatic representation of generating a Chinese restaurant process (CRP) using image data. At every sequence step $t$, a class is sampled according to a CRP and then an image from that class is sampled without replacement.

It can be shown (see Appendix A for details) that $x_d$ can be expressed as being drawn from exponential family with a Beta prior on the hurdle probability $p_d$ and a normal-inverse-gamma prior on the log-normal parameters $\mu_d$ and $\sigma_d^2$. We apply this exponential family model independently to each of the 512 dimensions. Since the optimal hyperparameters for the particle filter are *a priori* unknown, we metalearn the hyperparameters of the conjugate prior using the same minibatch setup as the neural circuit and Adam with learning rate of 0.1. Note that gradient estimation with respect to the hyperparameters is possible in the particle filter since the NLL of a sequence with sequential observations can be computed without the use of particles.

The results of this experiment are show in Table 2. Despite the effort to adapt the particle filter to this setting by carefully selecting the exponential family model, the neural circuit outperforms the particle filter by a large margin, both in terms of predictive performance and computational efficiency. Here, fully unobserved inference in the neural circuit is over $100\times$ faster than the particle filter, since exponential family inference in the particle filter must be performed separately over each of the 512 dimensions. As expected, predictive performance of the neural circuit drops when evaluating on novel classes drawn from the meta-test set, as these novel classes represent patterns of activations the circuit has not encountered during metalearning. However, the neural circuit still significantly outperforms the particle filter even in this difficult setting. Importantly, the architecture and training procedure of the neural circuit is identical to our setup in the experiment with synthetic data (Section 4.1), which speaks to the versatility of our method.

Table 2: Results on Imagenet-CRP with ResNet-18 activations as features. Evaluation computed as average over 10,000 held-out sequences of length 100. Negative log-likelihoods are expressed in nats per timestep.

|  | Meta-train Classes | | Meta-test Classes | | Inference Time (ms/sequence) | |
| Method | NLL ($\downarrow$) | ARI ($\uparrow$) | NLL ($\downarrow$) | ARI ($\uparrow$) | Seq. Obs. ($\downarrow$) | Fully Unobs. ($\downarrow$) |
|---|---|---|---|---|---|---|
| CRP | 1.005 | 0.010 | 1.003 | 0.009 | **0.019** | **0.185** |
| Softmax + Energy | 3.196 | 0.006 | 3.471 | 0.004 | 1883.066 | 1907.395 |
| Particle Filter | 0.848 | 0.070 | 0.933 | 0.048 | 2.407 | 73.896 |
| Neural Circuit | **0.255** | **0.749** | **0.680** | **0.271** | 0.067 | 0.412 |

### 4.3 Robustness to Distribution Shift: Open-set Classification on iNaturalist 2018

In order to test the robustness of our neural circuits to a strong form of distribution shift, we apply circuits trained using ImageNet-CRP to open-set classification on the long-tailed iNaturalist 2018 dataset [61]. iNaturalist consists of 437,513 images, each having labels at 7 taxonomic levels: kingdom, phylum, class, order, family, genus, and species. We therefore consider 7 versions of the task, each treating a different taxonomic level as the class label. This setting represents a distribution

shift not only in the images but also in the label statistics (though long-tailed, they are no longer drawn from a CRP).

Due to the fine-grained nature of iNaturalist 2018, for this section we modify the training of the neural circuits on ImageNet-CRP in several ways. First, we extlnd the sequence length to 500 during metalearning. The lower levels of the taxonomy have a large number of classes, meaning that the expected number of images per class is low for sequences of length 100. Second, we train a range of neural circuits, each with a different value of $\alpha \in \{1, 2, 5, 10, 20, 50, 100, 200\}$. This is because the coarser levels of the taxonomy may be better represented by low values of $\alpha$ and the finer levels better represented by large values of $\alpha$. Third, we use an ImageNet class split of 350 meta-train, 350 meta-validation, and 300 meta-test in order to mitigate the potential risk of overfitting.

Our baseline is a CRP with $\alpha$ tuned to provide the best performance on that task. For each taxonomic level, we evaluate the neural circuit trained with the $\alpha$ most similar to the CRP-tuned oracle $\alpha$. This is meant to emulate the setting in which a learner has only a rough estimate of the level of diversity expected when encountering a new environment. In order to help bridge the input distribution shift, we also apply an affine transformation and ReLU activation on top of the ResNet-18 features extracted from iNaturalist. The weights of this layer are trained using a small number ($n = 10$) of training sequences and early stopping is performed on the basis of validation sequences ($n = 10$). The weights of the neural circuit remain frozen. A neural circuit that performs better than the CRP means that transfer from ImageNet-CRP to iNaturalist has successfully occurred. We construct test sequences from iNaturalist by sampling 1,000 randomly permuted sequences of length 100.

Table 3: Dataset transfer from ImageNet-CRP to iNaturalist 2018. The $\alpha$ of the CRP baseline is tuned to provide optimal performance, whereas the neural circuit $\alpha$ is selected within $\{1, 2, 5, 10, 20, 50, 100, 200\}$ to be closest to the tuned CRP $\alpha$. Evaluation is performed on 1,000 randomly permuted sequences of length 100 from iNaturalist 2018 at each of 7 taxonomic levels. The average NLL per timestep is reported in terms of minimum, mean, and maximum across 5 runs with different random seeds. Best results are highlighted in bold.

| | CRP | | Neural Circuit (Ours) | | | |
|---|---|---|---|---|---|---|
| Taxonomy | Tuned $\alpha$ | NLL | Pretrained $\alpha$ | Min. NLL | Mean NLL | Max. NLL |
| Kingdom | 0.6 | 0.70 | 1 | **0.37** | **0.42** | **0.55** |
| Phylum | 2.1 | 1.30 | 2 | **0.80** | **0.84** | **0.88** |
| Class | 5.3 | 1.82 | 5 | **1.27** | **1.31** | **1.35** |
| Order | 33.1 | **2.24** | 20 | 2.15 | 2.28 | 2.49 |
| Family | 144.5 | **1.56** | 100 | 1.58 | 1.62 | 1.74 |
| Genus | 758.6 | **0.53** | 200 | 0.58 | 0.59 | 0.60 |
| Species | 1584.9 | **0.28** | 200 | 0.38 | 0.39 | 0.40 |

Our results (Table 3) show that positive transfer indeed occurs for three of the seven levels, indicating that the neural circuit is able to successfully transfer to novel open-set classification tasks in spite of distribution shift in both label and image statistics. Interestingly, the levels at which the neural circuit performs best are the higher levels, which there are likely to be multiple images per category.

We additionally perform an analysis to examine the effect of mismatch in $\alpha$ between meta-train and meta-test. Our results (see Appendix C.2) indicate that for low-moderate levels of $\alpha$, it may be beneficial to train on $\alpha$ slightly higher than anticipated, whereas for larger values of $\alpha$, it is best to match these statistics at meta-train time.

## 5   Related Work

Approximate solutions for nonparametric Bayesian models have historically used Markov chain Monte Carlo (MCMC) [44] or variational inference [64], with examples including Gaussian processes [60], Indian-Buffet processes [18, 59], and the Dirichlet process models we consider here [9, 47]. These approaches, however, assume a batch setting that is not efficient for sequential inference. Inference can be made more efficient in sequential settings by considering online variational inference with fixed update costs [10] or various forms of variational particle filtering [37, 42, 52]. However,

overly restrictive distributional assumptions (e.g. exponential family) can hinder performance when working with complex feature spaces such as neural network representations.

Amortized inference [35, 49] is better suited to complex problem domains since it learns a function that maps directly from inputs to an approximate posterior distribution, effectively amortizing the cost of variational inference. Several works have considered amortized inference over Dirichlet priors [21, 34, 43], Gaussian process priors [11], and nested Chinese Restaurant Process priors [26]. Similar to the classic approaches mentioned above, these methods belong to the family of batch inference solutions. Sequential variational autoencoders [15, 25, 28, 38] offer an online solution but none of these approaches are suitable for sequential inference in a DPMM.

Our neural circuit approach offers a scalable solution to sequential nonparametric Bayesian inference in a DPMM. We do so by using metalearning to incorporate the inductive bias of a nonparametric Bayesian model into the learned network rather explicitly instantiating such a model. This is reminiscent of previous metalearning approaches that train with "episodes" [63] to learn a general prior distribution over weight initializations [24, 27], languages [39], or supervised regression and classification problems [41]. Similarly, our method can be viewed as learning an implicit application of Bayes' rule in a sequential setting [5] where meta-learning defines Bayesian updates over complicated distributional assumptions [45]. We use sequences of observations and labels that can also be viewed as a kind of episode, but our goal is different: we aim to perform sequential inference over an unbounded number of classes. Compared to the previously mentioned approaches using variational inference or amortized inference, we leverage metalearning to directly learn a distribution over class labels rather than jointly learning encoder and decoder networks to maximize a variational lower bound on the log likelihood of the observations.

Related to our aims, the goal of open set recognition (OSR) is, broadly speaking, to detect previously unseen classes while accurately predicting known classes [54]. Several early approaches to OSR focus solely on detecting previously unseen classes, either using traditional machine learning methods [12, 13, 32, 55, 68] and, more recently, creating neural networks with open-set capabilities [7, 17, 57]. Bendale and Boult [6] first treated OSR as an incremental learning problem by using extracted image features to perform metric learning over known classes initially, and performing incremental class learning thereafter. This method employs thresholded distances from the nearest class mean as its decision function. Rudd et al. [50] advanced this approach by introducing distributional information into the thresholding process. Both methods rely on large datasets of known classes. [36] introduced a metalearning formulation of OSR based on thresholding prototypical embeddings [58] to address this limitation. Similarly, Willes et al. [67] proposed a method called FLOWR that combines prototypical embeddings with Bayesian nonparametric class priors. These methods operate in the sequential observation setting, observing the true class label after every prediction, and explicitly learn a metric space end-to-end instead of performing inference over arbitrary input features (including possibly representations extracted from a pretrained network) as we do. The particle filter baselines [23] we compare against closely resemble a variant of FLOWR modified for our setting.

# 6   Conclusion

We have proposed neural circuits for sequential nonparametric Bayesian inference: metalearning a recurrent neural network to capture the inductive bias of a DPMM that generates the training sequences. Our approach outperforms particle filter baselines in predictive performance while being more computationally efficient. Neural circuits are simple to implement and flexible enough to handle complex inputs with minimal modifications to their training procedure and architecture. In future work, we plan to use the neural circuit approach to capture inductive biases of more complex nonparametric Bayesian models with richer latent spaces.

Two limitations of our neural circuit approach are i) the difficulty of metalearning as $\alpha \to \infty$, and ii) misspecification of the base distribution when there is a mismatch between the data generating distribution of meta-train and meta-test sets. Classical DPMMs specify these quantities explicitly, making modeling assumptions easier to identify and reason about. The process of meta-learning – learning the model itself from data – brings about different challenges. Care should be taken when deploying the model in scenarios where the diversity in labels differs greatly from metalearning. Additionally, large $\alpha$ values may be difficult to learn due to the infrequency of repeated class instances

in meta-training sequences. In future work we plan to explore how these challenges can be addressed through curriculum learning.

Neural circuits are broadly applicable to the growing use of foundation models and pre-trained networks. They bridge the gap between Bayesian nonparametric methods, which often make more appropriate assumptions for real-world scenarios, and the powerful representations of neural networks. This integration allows existing neural networks to be adapted for tasks such as open-set recognition within a principled and efficient Bayesian framework, without the need to retrain the base model. We anticipate that this approach can be used in a wide range of settings to be able to deploy interpretable models with more explicit inductive biases build on top of expressive representations for complex datasets.

## Acknowledgments and Disclosure of Funding

This work was supported by grant N00014-23-1-2510 from the Office of Naval Research. The authors would like to thank Liyi Zhang for useful discussions.

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

# A Derivation of Hurdle Model

First we recall the form of an exponential family likelihood:
$$p(\mathbf{x} \mid \boldsymbol{\eta}) = h(\mathbf{x}) \exp \left\{ \langle \boldsymbol{\eta}, \mathbf{t}(\mathbf{x}) \rangle - A(\boldsymbol{\eta}) \right\}. \tag{13}$$
The corresponding conjugate prior takes the following form:
$$p(\boldsymbol{\eta} \mid \boldsymbol{\tau}, \nu) = \exp \left\{ \langle \boldsymbol{\tau}, \boldsymbol{\eta} \rangle - \nu A(\boldsymbol{\eta}) - B(\boldsymbol{\tau}, \nu) \right\}. \tag{14}$$

## A.1 Beta-Bernoulli

We first show how a Bernoulli likelihood and Beta conjugate prior can be expressed in terms of (13) and (14). This is a useful stepping stone towards the hurdle model, which contains the Bernoulli distribution as a component. Recall that for a Bernoulli likelihood $\text{Bern}(x|\theta)$,
$$p(x|\theta) = \theta^x (1-\theta)^{1-x} \tag{15}$$
$$\log p(x|\theta) = x \log \theta + (1-x) \log(1-\theta), \tag{16}$$
from which we recognize
$$\eta = \log \frac{\theta}{1-\theta} \tag{17}$$
$$t(x) = x$$
$$A(\eta) = -\log(1-\theta).$$
Similarly, recall that for a Beta conjugate prior $\text{Beta}(\theta|a,b)$,
$$p(\theta|a,b) = \frac{\Gamma(a+b)}{\Gamma(a)\Gamma(b)} \theta^{a-1}(1-\theta)^{b-1} \tag{18}$$
$$\tau = a - 1$$
$$\nu = a + b - 2$$
$$B(\tau, \nu) = \log \Gamma(a) + \log \Gamma(b) - \log \Gamma(a+b).$$

## A.2 Hurdle Model

Now suppose we have an arbitrary exponential family likelihood $p(x|\boldsymbol{\eta})$ of the form (13) and a conjugate prior $p(\boldsymbol{\eta}|\boldsymbol{\tau}, \nu)$ of the form (14). Now define a hurdle model based on this likelihood to be:
$$q(x|\tilde{\boldsymbol{\eta}}) = \begin{cases} 1-\theta & , x = 0 \\ \theta p(x|\boldsymbol{\eta}) & , x \neq 0. \end{cases} \tag{19}$$
Intuitively, this is an application of a Bernoulli variable that gates whether the underlying exponential family likelihood is active or not. Observe that $q(x|\tilde{\boldsymbol{\eta}})$ may be written as:
$$q(x|\tilde{\boldsymbol{\eta}}) = (1-\theta)^{(1-\mathbb{1}\{x \neq 0\})} (\theta h(x) \exp \left\{ \langle \boldsymbol{\eta}, \mathbf{t}(x) \rangle - A(\boldsymbol{\eta}) \right\})^{\mathbb{1}\{x \neq 0\}} \tag{20}$$

$$\log q(x|\tilde{\boldsymbol{\eta}}) = \log(1-\theta) + \mathbb{1}\{x \neq 0\} h(x) + \langle \boldsymbol{\eta}, \mathbf{t}(x)\mathbb{1}\{x \neq 0\} \rangle + \left( \log \frac{\theta}{1-\theta} - A(\boldsymbol{\eta}) \right) \mathbb{1}\{x \neq 0\} \tag{21}$$

We can therefore recognize $q(x|\tilde{\boldsymbol{\eta}})$ itself as exponential family:
$$q(x|\tilde{\boldsymbol{\eta}}) = \tilde{h}(x) \exp\{\langle \tilde{\boldsymbol{\eta}}, \tilde{\mathbf{t}}(x) \rangle - \tilde{A}(\tilde{\boldsymbol{\eta}})\} \tag{22}$$

$$\tilde{\boldsymbol{\eta}} = \begin{bmatrix} \log \frac{\theta}{1-\theta} - A(\boldsymbol{\eta}) \\ \eta_1 \\ \eta_2 \\ \vdots \end{bmatrix}$$

$$\tilde{\mathbf{t}}(x) = \begin{bmatrix} \mathbb{1}\{x \neq 0\} \\ t_1(x)\mathbb{1}\{x \neq 0\} \\ t_2(x)\mathbb{1}\{x \neq 0\} \\ \vdots \end{bmatrix}$$

$$\log \tilde{h}(x) = \mathbb{1}\{x \neq 0\} \log h(x)$$
$$\tilde{A}(\tilde{\boldsymbol{\eta}}) = -\log(1-\theta)$$

We also posit a conjugate prior for $q(x|\tilde{\boldsymbol{\eta}})$:

$$q(\tilde{\boldsymbol{\eta}}|\tilde{\boldsymbol{\tau}}, \tilde{\nu}) = \exp\{\langle \tilde{\boldsymbol{\tau}}, \tilde{\nu} \rangle - \tilde{\nu}\tilde{A}(\tilde{\boldsymbol{\eta}}) - \tilde{B}(\tilde{\boldsymbol{\tau}}, \tilde{\nu})\} \tag{23}$$

$$\log q(\tilde{\boldsymbol{\eta}}|\tilde{\tau}, \tilde{\nu}) = \tilde{\tau}_1 \tilde{\eta}_1 + \langle \boldsymbol{\tau}, \boldsymbol{\eta} \rangle - \tilde{\nu}\tilde{A}(\tilde{\boldsymbol{\eta}}) - \tilde{B}(\tilde{\boldsymbol{\tau}}, \tilde{\nu})$$

Now let $\mathrm{Bern}(x|\theta) = \bar{h}(x) \exp\{\langle \bar{\eta}, \bar{t}(x) \rangle - \bar{A}(\bar{\eta})\}$, with definitions following from (17). In this case, $\bar{\eta} = \log \frac{\theta}{1-\theta}$ and thus $\tilde{A}(\tilde{\boldsymbol{\eta}}) = \bar{A}(\bar{\eta})$. Moreover, $\tilde{\eta}_1 = \bar{\eta} - A(\eta)$, and we can then write

$$\log q(\bar{\eta}, \boldsymbol{\eta}|\tilde{\tau}, \tilde{\nu}) = \tilde{\tau}_1(\bar{\eta} - A(\boldsymbol{\eta})) + \langle \boldsymbol{\tau}, \boldsymbol{\eta} \rangle - \tilde{\nu}\bar{A}(\bar{\eta}) - \tilde{B}(\tilde{\tau}, \tilde{\nu}) \tag{24}$$

$$= \langle \boldsymbol{\tau}, \boldsymbol{\eta} \rangle - \tilde{\tau}_1 A(\boldsymbol{\eta}) + \tilde{\tau}_1\bar{\eta} - \tilde{\nu}\bar{A}(\bar{\eta}) - \tilde{B}(\tilde{\tau}, \tilde{\nu}),$$

from which we recognize that

$$\tilde{B}(\tilde{\tau}, \tilde{\nu}) = B(\tau, \nu = \tilde{\tau}_1) + \bar{B}(\tilde{\tau}_1, \tilde{\nu}). \tag{25}$$

Therefore, the hurdle applied on top of an arbitrary exponential family likelihood can be expressed in terms of the underlying base model and the Beta-Bernoulli presented in Section A.1.

## B  Softmax + Energy Model

The softmax + energy baseline is derived by viewing a standard logistic regression model through the lens of the maximum entropy principle [2]. Suppose that $\mathbf{w}_1, \ldots, \mathbf{w}_K$ are the vector of weights corresponding to each of $K$ known classes (it is assumed that the input $\mathbf{x}$ has been augmented with an dummy 1 feature to capture a bias term). The predictive distribution is then:

$$p(y = k|\mathbf{x}) = \frac{\exp(\mathbf{w}_k^\top \mathbf{x})}{\sum_{k'=1}^K \exp(\mathbf{w}_{k'}^\top \mathbf{x})} \tag{26}$$

This can be viewed as a maximum entropy distribution where each weight $w_{kd}$ is a Lagrange multiplier representing a constraint on $\mathbb{E}[\mathbb{I}[y = k]x_d]$ under the model. Now in order to allow for $y$ to take on the value $K + 1$, representing a new class, we introduce an additional constraint that the marginal probability that $y = K + 1$ follows the Chinese restaurant process $\beta = \frac{\alpha}{t-1+\alpha}$, where $t$ is the current timestep. The corresponding Lagrangian then becomes (discretizing $x$ and setting $D = 1$ for simplicity of illustration):

$$Q = -\sum_{k=1}^K \sum_i p_{ki} \log \frac{p_{ki}}{m_{ki}} + \lambda_0 \left(1 - \sum_{k=1}^{K+1} \sum_i p_{ki}\right) + \sum_{k=1}^K \lambda_k \left(\mu_k - \sum_i x_i p_{ki}\right) + \lambda_{K+1} \left(\beta - \sum_i p_{K+1,i}\right) \tag{27}$$

Setting $\frac{\partial Q}{\partial p_{ki}} = 0$ for $k = 1, \ldots, K$ and similarly for $\frac{\partial Q}{\partial p_{K+1,i}}$ yields:

$$p_{ki} = \frac{1}{Z} m_{ki} e^{-\lambda_k x_i} \tag{28}$$

$$p_{K+1,i} = \frac{1}{Z} m_{K+1,i} e^{-\lambda_{K+1}} \tag{29}$$

$$Z = \sum_i m_{K+1,i} e^{-\lambda_{K+1}} + \sum_{k=1}^K \sum_i m_{ki} e^{-\lambda_k x_i}. \tag{30}$$

Thus, we can compute the probability of a new class occurring as:

$$p(y = K + 1|x_i) = p_{K+1,i} = \frac{e^{-\lambda_{K+1}}}{e^{-\lambda_{K+1}} + \sum_{k=1}^K e^{-\lambda_k x_i}}, \tag{31}$$

which we recognize as a softmax with $w_k = -\lambda_k$ for $k = 1, \ldots, K$ and $-\lambda_{K+1}$ playing the role of an additional logit, which we will presently solve for. By setting $\frac{\partial Q}{\partial \lambda_{K+1}} = 0$, it can be shown that $e^{-\lambda_{K+1}} = \beta Z$. Substituting into (30) and solving for $Z$ yields:

$$Z = \frac{1}{1 - \beta} \sum_{k=1}^K \sum_i m_{ki} e^{-\lambda_k x_i}. \tag{32}$$

Therefore, we have that

$$-\lambda_{K+1} = \log\frac{\beta}{1-\beta} + \log\sum_{k=1}^{K}\sum_{i}m_{ki}e^{-\lambda_k x_i} \tag{33}$$

$$= \log\alpha - \log(t-1) + \log\sum_{k=1}^{K}\sum_{i}m_{ki}e^{w_k x_i}. \tag{34}$$

The derivation extends in the straightforward manner with $D > 1$. By considering infinitesimally small discretization and specifying a uniform measure $m(x)$ over $[-M, M]$, it can be shown that:

$$-\lambda_{K+1} = \log\alpha - \log(t-1) + \log\sum_{k=1}^{K}\exp\left(\sum_{d=1}^{D}\log\frac{\sinh(Mw_{kd})}{Mw_{kd}}\right). \tag{35}$$

Similarly, choosing a Gaussian measure $m(x) = \exp\left(-\frac{x^2}{2\sigma^2}\right)$ produces

$$-\lambda_{K+1} = \log\alpha - \log(t-1) + \log\sum_{k=1}^{K}\exp\left(\sum_{d=1}^{D}\left[\frac{1}{2}\log(2\pi) + \log\sigma + \frac{\sigma^2 w_{kd}^2}{2}\right]\right). \tag{36}$$

We use (35) as the augmented new class logit in the ImageNet experiments as this better represents the state of knowledge for the ResNet-18 activations and similarly we use (36) for the 2-D synthetic DPMM augmented logits. We learn the weights of the logistic regression classifier after every timestep by using gradient descent with the Adam [3] optimizer.

## C  Additional Experimental Results

In this section, we include additional experimental results.

### C.1  Visualization of Predictive Distribution

In Figure 4, we visualize the evolution of the neural circuit predictive distribution across timesteps.

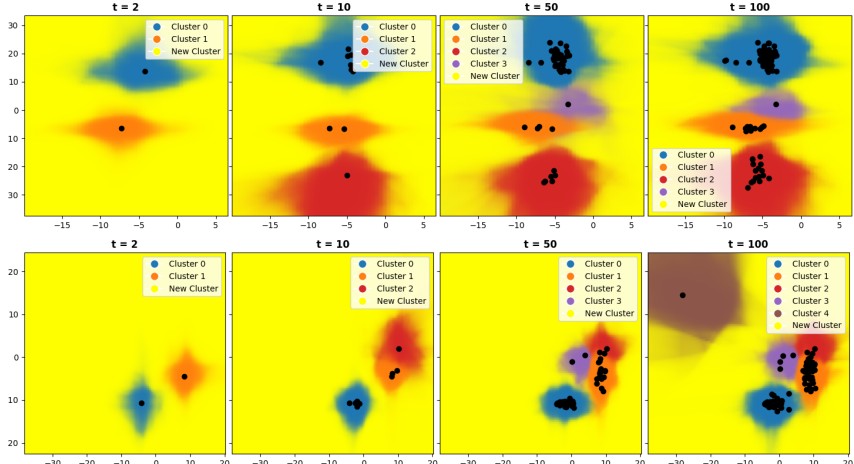

Figure 4: Visualization of the predictive distribution over cluster assignments across timesteps for a neural circuit model trained on the two-dimensional data synthesized from a DPMM. Colors are determined by mixing each cluster's color proportional to the output softmax probabilities. Yellow indicates a new, as yet unobserved class.

## C.2   Effect of Varying $\alpha$

In this section, we evaluate the effect of a mismatch in $\alpha$ between meta-train and meta-test. We take the neural circuits trained on ImageNet-CRP from Section 4.3 and evaluate all possible combinations of meta-train and meta-test $\alpha$. The results (Table 4) show that for low-moderate levels of $\alpha$, it may be beneficial to train on $\alpha$ slightly higher than anticipated. This is likely due to their increased level of diversity. However, when evaluating on large values of $\alpha$, it tends to be best to match these statistics at meta-train time.

Table 4: NLL on ImageNet-CRP Meta-test Classes under different meta-training conditions. Evaluation is performed on 1,000 sequences of length 500.

| Meta-Test $\alpha$ | Meta-Training $\alpha$ | | | | | | | |
|---|---|---|---|---|---|---|---|---|
| | 1 | 2 | 5 | 10 | 20 | 50 | 100 | 200 |
| 1 | 0.76 | **0.75** | 0.76 | 0.78 | 0.87 | 1.10 | 1.25 | 2.30 |
| 2 | 1.24 | 1.19 | **1.18** | 1.21 | 1.31 | 1.51 | 1.83 | 2.80 |
| 5 | 2.26 | 2.02 | 1.90 | **1.89** | 1.97 | 2.16 | 2.41 | 3.09 |
| 10 | 3.37 | 2.94 | 2.62 | **2.52** | 2.53 | 2.67 | 2.88 | 3.30 |
| 20 | 4.13 | 3.78 | 3.40 | 3.23 | **3.06** | 3.08 | 3.26 | 3.56 |
| 50 | 4.60 | 4.44 | 4.22 | 4.58 | 3.83 | **3.49** | 3.55 | 3.77 |
| 100 | 4.79 | 4.74 | 4.63 | 5.06 | 4.34 | 3.97 | **3.47** | 3.53 |
| 200 | 4.94 | 4.94 | 4.91 | 5.37 | 4.69 | 4.51 | 3.45 | **3.08** |

## C.3   Additional Results on Synthetic DPMM Data

In Table 5, we show additional metrics collected in the synthetic DPMM setting from Table 1. Here we show the results in terms of min, mean, and max across 5 training runs with different random initializations.

Table 5: Results on two-dimensional data synthesized from a DPMM for 5 neural circuits each trained with a different random initialization. Results are summarized as the min, mean, and max of the 5 models. Inference times are reported in units of ms/sequence.

| Metric | Min. | Mean | Max. |
|---|---|---|---|
| NLL ($\downarrow$) | 0.075 | 0.076 | 0.078 |
| Perplexity ($\downarrow$) | 1.085 | 1.086 | 1.089 |
| ARI ($\uparrow$) | 0.920 | 0.921 | 0.922 |
| AMI ($\uparrow$) | 0.927 | 0.928 | 0.929 |
| Inf. Time (Seq. Obs.) ($\downarrow$) | 0.056 | 0.059 | 0.064 |
| Inf. Time (Fully Unobs.) ($\downarrow$) | 0.394 | 0.421 | 0.445 |

## C.4   Additional Results on ImageNet-CRP Data

Similarly, we show in Table 6 the min, mean, and max results of neural circuits trained with 5 different random initializations.

# D   Additional Experimental Details

For all of our experiments neural circuit RNN cell was chosen to be a 2-layer gated recurrent unit (GRU) [1] with hidden size 1024. The maximum number of output logits was set to be equal to the sequence length used for metalearning, as this is the maximum number of classes that could possibly be encountered. Training was performed over 10,000 minibatches each of size 128 sequences of length 100 (256 sequences of length 500 for the iNaturalist experiments). The CRP coefficient was set to $\alpha = 1.0$ for the synthetic and ImageNet-CRP experiments, and was set to a range of different values $\{1, 2, 5, 10, 20, 50, 100, 200\}$ in the iNaturalist experiments. The neural circuit was trained

Table 6: Results on ImageNet-CRP for 5 neural circuits each trained with a different random initialization. Results are summarized as the min, mean, and max of the 5 models. Inference times are reported in units of ms/sequence.

| Metric | Min. | Mean | Max. |
|---|---|---|---|
| Meta-train NLL ($\downarrow$) | 0.239 | 0.255 | 0.272 |
| Meta-train ARI ($\uparrow$) | 0.728 | 0.749 | 0.770 |
| Meta-test NLL ($\downarrow$) | 0.668 | 0.680 | 0.696 |
| Meta-test ARI ($\uparrow$) | 0.260 | 0.271 | 0.286 |
| Inf. Time (Seq. Obs.) ($\downarrow$) | 0.066 | 0.067 | 0.068 |
| Inf. Time (Fully Unobs.) ($\downarrow$) | 0.407 | 0.412 | 0.420 |

using Adam with learning rate $0.001$ and the particle filter using learning rate $0.1$. For the synthetic and ImageNet-CRP results, the tuning hyperparameters (e.g. model architecture, learning rates) were selected by monitoring training loss over the first several hundred minibatches and modifying as necessary. For the iNaturalist 2018 results, overfitting was monitored on a separate meta-validation class split. MAP prediction is implemented in the neural circuit by simply selecting the class label with highest predicted probability for each timestep.

One random seed was used for training each method. The clustering metrics were computed using the `adjusted_rand_score` and `adjusted_mutual_info_score` functions from Scikit Learn [5]. Experiments were performed using NVIDIA A100 GPUs with 40 GB of GPU memory and 64 GB of CPU memory across 4 threads. Training of the neural circuits took approximately 6-8 hours per run.

For the particle filter, we use 100 particles for inference and utilize an adaptive resampler that resamples according to a multinomial distribution over the particles whenever the effective sample size drops below 50. In the fully unobserved setting, we take the MAP prediction to be the particle with the largest weight after running the filter on a sequence.

For the experiments in Section 4.2, the meta-training classes were taken to be the first 500 and the meta-test classes were taken to be the last 500 classes, as sorted by class ID. For Section 4.3, the meta-training classes were the first 350 classes, the meta-validation classes the next 350 classes, and the meta-test classes the final 300 classes, as sorted by class ID.

All methods are implemented in PyTorch [4] and are GPU-enabled. Our code is available online [1].

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
