# OpenReview forum: "A Metalearned Neural Circuit for Nonparametric Bayesian Inference"
_NeurIPS.cc/2024/Conference — NeurIPS 2024 poster_

### Official Review · Reviewer_SHmK · 2024-06-25

**Soundness:** 3
**Presentation:** 3
**Contribution:** 2
**Rating:** 6
**Confidence:** 3

**Summary:**

The authors present a recurrent neural net (RNN) that learns to mimic a Bayesian non-parametric (BNP) approach to classification with (potentially) heavy-tailed data and an a priori unknown number of classes. The method is evaluated on three experimental setups.

**Strengths:**

The paper itself is well-written and motivated. It provides a clever approach to combining BNP with the straightforwardness of an RNN.
Conditioned on the experimental weakness mentioned below, the improvements seem significant. However, as I am not familiar with the literature I cannot properly judge the significance, as I am not sure (but somewhat doubtful) whether the state of the art on this problem has not progressed in the last twenty years as the chosen baselines.

**Weaknesses:**

The weaknesses lie primarily in the experimental evaluation.
- As discussed, the only baselines are a publication by Fearnhead from 2004, a vanilla Chinese restaurant process, and a self-designed non-Bayesian model. This hardly provides a proper evaluation of the approach.
- All results are reported from a single seed without any significance analysis. (And no, promising multiple runs in a future revision of the paper is not sufficient to answer _yes_ to this question in the checklist. _No_ would have been the correct answer.)

## Minor
- The abstract claims better performance than _"particle filter-based method**s**"_ while only a single method is being compared against
- Given that there is a single seed, reporting four decimals of significance seems unjustified.

**Questions:**

- Only a single run is reported for each result. Was the seed to generate the same for all methods? E.g., has every model in Sec 4.1 access to the same training data?
- In sec 4.2 the choice is to pick the representation space post-relu activations, placing an extra hurdle on the particle filter baselines (the hurdle model). Why was this choice made instead of sticking to common practice for taking the pre-activation representation features from pre-trained neural nets?  Which outcome would the authors expect without this artificial introduction of sparsity?

**Limitations:**

The limitations of the approach are adequately discussed by the authors.

---

> ### Author Rebuttal · Authors · 2024-08-07
>
> We thank the reviewer for their constructive comments. Regarding reproducibility, please see the attached PDF (under general author rebuttal), where we have added results for neural circuits trained using 5 different random initializations. Please note that the variation in results across these random seeds is low and the paper will be updated to reflect this measure of uncertainty. Each model has access to the exactly the same training data and is evaluated on identical precomputed test sequences. Thank you also for the suggestion to add more baselines. We will update the paper to add these. In terms of activations, our method is agnostic to the input representation. We chose to use post-relu activations as this is analogous to the final projection layer of standard neural networks. However, we will also try the pre-relu activations and update the paper accordingly.

---

> > ### Comment · Reviewer_SHmK · 2024-08-08
> >
> > Thank you for your answer and these additional results

---

### Official Review · Reviewer_aqs6 · 2024-07-04

**Soundness:** 2
**Presentation:** 3
**Contribution:** 3
**Rating:** 6
**Confidence:** 4

**Summary:**

This paper proposes a novel method for inference for Bayesian nonparametric models through an amortized approach. After simulating data from a Dirichlet Process Mixture Model, a Recurrent Neural Net is trained to characterise the relationship between the simulated data and the parameters, and consequently characterise the posterior distribution. Comparisons are made with a particle filter method for sequential inference for a DPMM, and experiments are run on simulated data, and image classification tasks using pre-trained neural net activations as inputs.

**Strengths:**

This is an interesting article taking a novel approach to inference for DPMMs. The motivation and theory are adequately explained, and the application examples chosen are interesting. The use of amortized inference in this context is novel: there are some interesting links to make here with Likelihood Free Inference (LFI) methods, where using a neural net to model the relationship between simulated data and the parameter space is becoming more common. See https://www.pnas.org/doi/10.1073/pnas.1912789117 for a relevant review. The quality of the scientific writing is in general high and concepts are explained clearly.

**Weaknesses:**

The experiments are not thorough: they each use one random seed per method, which gives no indication as to the uncertainty or robustness of the results. This is especially confusing, since the authors make a point of describing the new method as being faster than existing methods.

The computational resources needed for training the neural net are quite expensive, so maybe tone down the enthusiasm about this method being so computationally efficient. At least make clear that this method makes effective use of a generous budget of compute resources.

The examples used are interesting but quite limited in scope: the first example is (reasonably enough) a simple mixture of Gaussians, and the subsequent examples are practical but involve somewhat esoteric knowledge of using pre-trained neural nets as training data and similar. The article would benefit from a clustering example based on real observational data, as hinted at in the introduction. This would make the methods and article much more approachable for a more general audience

Line 17/57: I wouldn’t call a mixture model a “classification model” exactly: that word is normally associated with supervised learning and known discrete labels. This is more generally considered “clustering”

Line 76: “Chinese restaurant process” might need a little explaining for people who haven’t heard the analogy before. Otherwise good!

**Questions:**

Can this method be adapted for non-streaming data? Would a more general class of neural network architecture work for a different data paradigm?

Have you considered a comparison with finite parametric mixture models, or overfitted finite mixtures? These are being given greater consideration now after some inconsistency results for DP clustering models, and the inference is substantially easier.


Would this inference method work with a finite parametric mixture model, or a truncated nonparametric model?

High dimensional (in the observed covariate space) model-based clustering is in general hard. Would this method have an advantage here, as neural nets normally adapt pretty well to high-dimensional spaces?

Line 32: “nonparametric Bayesian models have fallen out of favour in machine learning” Really? Give more justification to this statement, or provide some examples of how they have been replaced. If they have fallen out of favour, why are you working with them? Is it just the inference that is holding them back from flourishing, or is it something else?

Line 203: For a more general audience: What exactly is a ResNet? What has it been trained on? What kind of properties do you expect its activations to have relevant to the task? Can you explain this more thoroughly before going further?

Line 227/228: where are the computational efficiency results exactly? They are not visible where you mentioned them.

Line 248: is there no way of tuning the alpha within the neural net framework? If not, please explain why. The additional analysis looking at the transferability of alpha (line 262) is good, however!

Line 272: Can you explain more clearly why none of references [10,37,42,52] are appropriate comparisons? Is there nothing more usable than the fiddly particle filter model?

Can you do better than MAP prediction? What kind of posterior uncertainty does this method provide, if any?

Line 551: “One random seed was used for training each method”? Really? Is this why there is no errors on the results? If so, this is a major weakness of the results here hidden in the Appendix. From the code of conduct, it seems that this is the case, and will be “updated to multiple runs in the next version of the paper” - does that include the revised version for Neurips?

Line 553: “Experiments were performed using NVIDIA A100 GPUs” why the plural here? How many GPUs per experiment? This is unclear and an important detail.

**Limitations:**

Table 1/2/3/4: there is no uncertainty/errors on the losses. This is because everything was done with a single random seed, so maybe you’re not in a place to add uncertainty easily.

Line 227-228: claims are made about the computational efficiency of the method that aren’t really backed up, and are possibly undermined by the inability to use more than one random seed.

---

> ### Author Rebuttal · Authors · 2024-08-07
>
> We thank the reviewer for their detailed and constructive comments. In particular, we are grateful for the suggestion to improve our analysis by training models with multiple random initializations. Please refer to the PDF under the general author rebuttal above, where we show the results of training 5 different random initializations. These results show that there is relatively little variation across different runs of the model. We will update the paper to include these measures of uncertainty.
>
> We also thank the reviewer for raising the point about the computational cost of training the network. Our perspective had been focused primarily on the situation at inference time after the models have already been trained. However, the reviewer is quite right that there are applications where a trained network is used for inference only a few times, in which case the amount of computation required for training may become burdensome. We will add discussion to the paper about this point and adjust our claims regarding computational efficiency to reflect this.
>
> The suggestion to consider a clustering example based on real observational data is a welcome one. As our method can be applied to any input modality where a deep neural network is able to provide meaningful representations, we anticipate that our method would also perform well in this setting. We will add experiments to demonstrate this.
>
> In terms of non-streaming data, our method could be adapted to this scenario by assigning an artificial order, or by sampling several random permutations and averaging over the predictions. Similarly, our method is compatible with any distribution defined over the cluster assignments, including finite and truncated nonparametric distributions. The only requirement is that it is tractable to draw samples from this distribution. In preliminary experiments, we had tried a uniform distribution over a fixed number of K classes, and found that the neural circuit was able to handle this case as well. We will add a section to the experiments that explores this setting.
>
> Our hope is that by focusing directly on the mapping from input to cluster assignment, our proposed method can better handle high-dimensional input spaces. This may be contrasted with particle filters, where an explicit cluster conditional distribution must be posited in the input space. This may be relatively straightforward in lower dimensions but for higher dimensions it becomes difficult to specify a suitable distribution. Neural circuits circumvent this issue to some degree by learning to directly predict a distribution over clusters. We invite the reviewer to please refer to the attached PDF under the general author rebuttal, which visualizes the predictions in the two-dimensional setting. It can be observed that the neural circuit is somewhat more flexible than a Gaussian cluster-conditional distribution and can instead learn an appropriate mapping to the cluster assignments directly.
>
> Regarding nonparametric Bayes, our intent was not to disparage NPB. On the contrary, our work is aimed at marrying the unique advantages of NPB with deep neural networks. We meant to say that large-scale deep models have become relatively more popular than nonparametric Bayes in recent years. We will update the paper to make this point clearer.
>
> We will explain more clearly our use of ResNets as a representation, briefly stating that they are a popular choice for image classification and it is straightforward to download and use pretrained ResNets. The cost of training a large-scale image classifier from scratch can be prohibitive, so we see the ability to perform inference over fixed embeddings as a strength.
>
> The choice of alpha can be viewed as a modeling assumption that reflects any knowledge that the practitioner has regarding the rough growth rate of classes. Since our method utilizes metalearning, there is a great deal of flexibility here. For example, if a practitioner is unsure about the exact sort of growth rate but is confident that the alpha will be in the interval [1, 10], then e.g. a uniform distribution could be placed on alpha and used when sampling the minibatches for training. The resulting neural circuit would thus be trained to be robust to selection of alpha within this range, perhaps at the cost of performance for a specific alpha.
>
> Thank you for the suggestion to add more particle filter baselines, including those with a fixed number of clusters and variational approaches. We will update the paper to add these baselines.
>
> In terms of posterior uncertainty, posterior cluster assignments may be sampled from the neural circuit and then used to estimate statistics as appropriate. We chose to use MAP prediction for the sake of simplicity in our fully unobserved evaluations as it should represent the model’s best single guess about the cluster assignment.
>
> The connection to likelihood-free (simulation-based) inference is highly relevant as the CRP sampler plays this role by selecting the sequence of cluster assignments to be used in each minibatch of training. We will add discussion about this to the related work section.
>
> We thank the reviewer again for their many insightful comments, which have greatly improved our paper. We hope that the reviewer would consider increasing their score based on these changes.

---

> > ### Comment · Reviewer_aqs6 · 2024-08-12
> >
> > I appreciate the edits made by the authors and the level of engagement shown.
> >
> > I am particularly appreciative of the more thorough experimental results with comparisons to other methods and the use of multiple random seeds.
> >
> > I am therefore happy to increase my score.

---

### Official Review · Reviewer_QfVx · 2024-07-10

**Soundness:** 2
**Presentation:** 2
**Contribution:** 2
**Rating:** 6
**Confidence:** 3

**Summary:**

This paper presents an approach for nonparametric Bayesian inference using metalearning to train a recurrent neural network (RNN) to perform sequential inference over an unbounded number of classes. The key contributions are: (a) method to extract the inductive bias from a Dirichlet process mixture model (DPMM) and transfer it to an RNN through metalearning, (b) "neural circuit" architecture that can perform fast sequential inference over an open set of classes after training and (c) experimental results showing the neural circuit achieves comparable or better performance than particle filter methods while being faster and simpler to use. The proposed approach consists of first generating training sequences from a DPMM prior. These sequences are then used to train an RNN to predict class labels sequentially. When trained to completion the RNN captures the underlying structure from the sequences (DPMM) and consequently can be used for fast inference. The authors include empirical results on synthetic data from a known DPMM, as well as more realistic experiments on ImageNet features with classes sampled from a Chinese Restaurant Process and a long-tailed iNaturalist species classification dataset. The neural circuit outperforms particle filter baselines in predictive performance and computational efficiency across these tasks.

**Strengths:**

* The proposed method is a neat way of combining the strengths of nonparametric Bayesian models and deep neural nets. By using a RNN to amortize inference in a DPMM, the paper leverages the flexibility of DPMM and RNNs. The paper falls in a line of recent work using NNs to amortize Bayesian inference, e.g. [1]. So while the idea of approximating inference with NNs is not completely new, this paper is the first to study it in the context of DPMMs.
* While not perfect as I will discuss below, the empirical analysis is quite thorough. The authors present experiments on a relatively simple synthetic domain as well as more challenging image-based experiments. The amortized approach perform better than or at least on par with the particle filtering baseline while being much faster at inference.
* The method also appears to demonstrate some robustness to distribution shifts as the experiment of tranferring the ImageNet-trained model to iNaturalist shows.
* The authors also provide code to aid reproducibility.

[1] Transformers Can Do Bayesian Inference. Samuel Müller, Noah Hollmann, Sebastian Pineda Arango, Josif Grabocka, Frank Hutter. ICLR 2022.

**Weaknesses:**

* While the results are certainly significant, I find the experimental analysis lacking in several aspects. Firstly, the experiments in the more realistic open set image classification task are limited by fixing a pretrained featurizer to embed the image into a lower dimensional vector. This removes significant complexity from the inference task and also means the method already has access to some information about the underlying inference task. This also makes the experiments of transfer to the iNaturalist task a bit less impressive since the ResNet featurizer already embeds the images in a "similar" space.
* The authors also perform all the experiments using a relatively small RNN (2-layer GRU). There isn't much of a discussion on how this particular architecture was chosen and more importantly there are no experiments to understand the effect of this choice. As this is a largely empirical paper, it is important to study these design choices. For instance, how does the performance vary if we use a larger RNN or if we use a transformer instead? Other such design choices are also not accounted for.
* The comparison is somewhat limited to particle filter baseline, and despite the focus on open-set image classification, there is no consideration for other standard baselines for the task [2].
* Another limitation of the approach is robustness to model misspecifcation. Distilling the structure of a (fixed) DPMM into a RNN can make it harder to generalize to data coming from different (unknown) data generating process.

[2] Large-Scale Open-Set Classification Protocols for ImageNet. Andres Palechor, Annesha Bhoumik, Manuel Günther. WACV 2023.

**Questions:**

Please see weaknesses section above.

**Limitations:**

The authors discuss the limitations of the method in the conclusion, but I would also encourage them to discuss some of the aspects I highlight in the Weaknesses section.

---

> ### Author Rebuttal · Authors · 2024-08-07
>
> We thank the reviewer for their detailed and constructive feedback. The reviewer is correct that in our experiments we assume the use of a pretrained featurizer. Our initial motivation was to demonstrate that neural networks originally trained on a fixed number of classes can easily be adapted to the open-set setting by replacing the penultimate layer with our meta-trained circuit. Backpropagating through the pretrained feature extractor or training it directly from scratch are interesting avenues to explore and we are grateful to the reviewer for raising this point.
>
> Regarding design decisions of the network, we have run preliminary experiments exploring the role of hidden size in [512, 1024, 2048] and number of RNN layers in [1, 2, 3]. We trained these using the class split discussed in section 4.3 and found that all models had a similar final validation loss, with each falling in the range [0.7582, 0.7791]. Thus the performance is not very sensitive to this choice. We will add this ablation to the paper.
>
> We also thank the reviewer for suggesting other baseline methods in the open-set literature. The primary barrier of previous open-set approaches is that they typically assume a considerable amount of training data for each of the known classes so that e.g. a prototype representation can be estimated for each class and distances computed to these in order to estimate the probability of a new class. This is problematic in the settings we consider here, as there may be only one or two class instances available for several of the classes. We thank the reviewer for the reference and will add another suitable open-set baseline to the paper accordingly.
>
> Regarding robustness to model misspecification, it is true that our method may struggle if the data-generating process is significantly different from that during training. We have explored this in Appendix C, Table 4 of the paper which provides some guidelines on choosing a good value of alpha that is able to handle variations to this at test time. This is a drawback shared by particle filters as well, which require specifying a prior assumption over alpha. However, one advantage of our proposed method is that in principle the metalearning procedure could incorporate perturbations to the alpha used to construct training minibatches, whereas it is not straightforward to adapt particle filters similarly. We will add more discussion about this point to the paper.
>
> We thank the reviewer again for their insightful comments which have greatly strengthened our paper.

---

> > ### Comment · Reviewer_QfVx · 2024-08-08
> > **Response to rebuttal**
> >
> > Thanks for the response and additional experiments which are helpful. I will maintain my score.

---

### Official Review · Reviewer_grnU · 2024-07-13

**Soundness:** 3
**Presentation:** 3
**Contribution:** 3
**Rating:** 6
**Confidence:** 4

**Summary:**

This paper proposes to use recurrent neural network (RNN) for classification with an open (non-fixed) number of class labels. This is motivated by non-parametric Bayesian models such as Dirichlet process mixture model and the iterative update on exponential family sufficient statistics in its particle filter methods. The latent state of RNN would resemble the iteratively updated sufficient statistic, and can be used for classifying data into previously seen or a first time occurring class label. Experiments on synthetic data and image data are provided, along with an extended study on robustness to distribution shift.

**Strengths:**

1. The paper is well organized and written, with enough background and clear motivation.
2. The proposed method is interesting and innovative, bridging the two largely non-overlapping fields of Bayesian nonparametrics and deep learning.

**Weaknesses:**

1. It seems that the proposed method only does the classification task, but looses the inference side of Bayesian nonparametric methods.
2. The ability to assign label to a class occurring for the first time is through the mask m_t in Eq.(10), whereas the dimensions of weight matrix W, mask m_t have to be pre-fixed. That is, we are not really assuming there can be infinitely many classes as in Bayesian nonparametric methods, but instead picking a large enough number and assuming that the number of classes is smaller (this is not necessarily a weakness, just some differences I noticed).

**Questions:**

1. Have you tried visualizing the learned weight matrix W? It would be interesting to see if the probability of introducing a new cluster decays with the number of data and the number of observed clusters like Chinese restaurant process.
2. How does the expected number of clusters scale with the number of data? Does the curve resemble some Pitman-Yor process?
3. How could I use this approach as an efficient approximation of Bayesian nonparametric methods and make inference (e.g. getting posterior distributions or some summary statistics of it)? I could imagine that we repeatedly draw random samples from the softmax probabilities in Eq.(10) instead of taking argmax, but this would just become another version of particle filter methods. Is there some better ways of doing this? If so, then weakness 1 wouldn't really be a weakness.

**Limitations:**

See weaknesses and questions.

---

> ### Author Rebuttal · Authors · 2024-08-07
>
> We thank the reviewer for their constructive comments. Visualizing the weight matrix W and examining the relationship between the expected number of clusters and the number of observations are both excellent suggestions. The probability of predicting a new cluster can be estimated in the synthetic data case via Monte Carlo sampling, as the marginal distribution in input space is known. We will update the paper to add this analysis.
>
> Regarding posterior inference, our method is primarily intended to produce a posterior distribution over cluster assignments. Although this does not provide sufficient statistics directly, from the visualizations in the above PDF (see the general rebuttal above), it is evident that this information is being represented in the hidden state of the RNN as it adapts over time to accommodate new clusters. Posterior per-cluster statistics could be estimated by sampling cluster assignments from the model and then averaging the statistic over observations assigned to the same cluster. We will update the paper to add discussion about this point.
>
> The reviewer is correct in that the proposed method is not truly infinite in that there is an upper bound on the number of classes, determined by the output layer of the RNN. Care should be taken to choose an appropriate maximum output size with respect to the anticipated sequence length and growth rate of classes. As CRPs have a logarithmic relationship between the number of observations and the number of occupied clusters, we do not view this as a major drawback, but we will nevertheless make this point clearer by adding it as a limitation in the discussion.
>
> We thank the reviewer for their insightful questions which have helped to improve our paper.

---

> > ### Comment · Reviewer_grnU · 2024-08-12
> >
> > Thank you to the authors for the detailed reply. Most of my concerns and questions have been addressed, and I will increase my rating accordingly.

---

### Author Rebuttal · Authors · 2024-08-07

We thank each of the reviewers for their feedback. In response to the suggestions provided by the reviewers, we have prepared several new results in the attached PDF.

We appreciate the suggestion to improve intuitions about the approach through visualizations. We have added plots visualizing the predictions for synthetic data, in which it is possible to clearly see the predictions of unobserved classes.

We have also added additional results to support the reproducibility of our findings. We have added results with five random seeds, and show that our findings hold up across this sample. We also note that each method has access to the same training and test data, and clarify that for each run both training and evaluation was performed using a single A100 with 80GB of GPU memory. Please note that the inference times have improved somewhat since the previous neural circuit inference times had been inadvertently collected on a less capable GPU, and this has now been remedied.

We hope the reviewers find these additions to our results convincing, and we appreciate the time and effort that went into generating such constructive feedback. We provide a detailed response to the points raised by each reviewer in the individual responses below.

---

### Comment · Area_Chair_tZHb · 2024-08-09

The authors-reviewers discussion period has now started.

@Reviewers: Please read the authors' response, ask any further questions you may have or at least acknowledge that you have read the response. Consider updating your review and your score when appropriate. Please try to limit borderline cases (scores 4 or 5) to a minimum. Ponder whether the community would benefit from the paper being published, in which case you should lean towards accepting it. If you believe the paper is not ready in its current form or won't be ready after the minor revisions proposed by the authors, then lean towards rejection.

@Authors: Please keep your answers as clear and concise as possible.

The AC

---

### Decision · Program_Chairs · 2024-09-25

**Decision:**

Accept (poster)

**Comment:**

This paper proposes a novel method for performing inference with Bayesian nonparametric models, specifically Dirichlet Process Mixture Models (DPMMs), by leveraging recurrent neural networks (RNNs). The authors achieve this by first simulating data from a DPMM and then training an RNN to classify this data. The trained RNN, called a neural circuit, can then be used to perform inference over an unseen sequence of data, potentially containing new classes not observed during training. The paper presents the strengths of this method to be its ability to handle open-set classification tasks (classifying data with potentially new classes) and its efficiency compared to traditional particle filter methods for BNP inference.

Strengths:

The paper is well-written and clearly explains the proposed method, its motivation, and its background.
The idea of combining Bayesian nonparametric models with neural networks is original and with potential for impact.
The paper presents experiments on synthetic data and real-world image classification tasks demonstrating the effectiveness of the proposed method.
The method offers advantages in terms of efficiency compared to traditional particle filter methods for BNP inference.

Weaknesses:

The reviewers raised concerns about the limited scope of the experiments, particularly the use of a pre-trained featurizer on the image classification tasks.
The lack of exploration of different network architectures and hyperparameters was also noted by some reviewers.
The paper could benefit from a more thorough comparison with existing open-set classification methods.
The limitations of the method, such as its potential sensitivity to model misspecification, should be discussed in more detail.

Recommendation:

All the reviewers voted for acceptance. I therefore recommend acceptance and encourage the authors to use the feedback provided to improve the paper for its final version.